# Diagnosis and Intervention in Early Psoriatic Arthritis

**DOI:** 10.3390/jcm11072051

**Published:** 2022-04-06

**Authors:** Tomoyuki Hioki, Mayumi Komine, Mamitaro Ohtsuki

**Affiliations:** 1Department of Dermatology, Central Japan International Medical Center, Minokamo 505-8510, Japan; 2Department of Dermatology, Jichi Medical University, 3311-1 Yakushiji, Shimotsuke 329-0498, Japan; mamitaro@jichi.ac.jp

**Keywords:** psoriatic arthritis, early diagnosis, treatment, early intervention

## Abstract

Psoriatic arthritis (PsA) is a chronic inflammatory disorder that affects approximately 20–30% of patients with psoriasis. PsA causes deformities and joint damage, impairing quality of life and causing long-term functional disability. Several recent studies demonstrated that early diagnosis and intervention for PsA prevents permanent invalidity. However, the clinical features of PsA vary and are shared with other differential diseases, such as reactive arthritis, osteoarthritis, and ankylosing spondylitis. The common and overlapping features among these diseases complicate the accurate early diagnosis and intervention of PsA. Therefore, this review focuses on the current knowledge of the diagnosis of early PsA and discusses the meaning of early intervention for early PsA.

## 1. Introduction

Psoriatic arthritis (PsA) is a chronic inflammatory disorder that affects 14.0–22.7% of patients with psoriasis [1,2,3]. The incidence of PsA differs among counties: 22.7% in European psoriasis patients, 21.5% in South American patients, 19.5% in North American patients, 15.5% in African patients, and 14.0% in Asian patients with psoriasis [3]. A Japanese Society for Psoriatic Research survey revealed a 10.5% occurrence of PsA among newly visited psoriasis patients [4]. Its prevalence varies from 0.19% to 0.25% [5,6]. The main musculoskeletal manifestations of PsA are peripheral arthritis, spinal spondylitis, asymmetrical synovitis, enthesitis, and/or dactylitis [7]. In 1973, Moll and Wright proposed classifying PsA into five subgroups: (1) asymmetric oligoarthritis, (2) predominant distal interphalangeal joint involvement, (3) symmetric polyarthritis, (4) predominant axial involvement, and (5) arthritis mutilans [8].

PsA is a highly heterogeneous disease whose clinical features vary [9]. Its clinical features are also observed in other diseases, such as reactive arthritis, osteoarthritis, and ankylosing spondylitis [10]. The common and overlapping features of these diseases present challenges in the accurate diagnosis of PsA. The delayed diagnosis of PsA is associated with poor physical function and permanent invalidity [11,12]. There is increasing concern that early diagnosis and a rapid therapeutic intervention, such as biologics before the onset of structural damage, can inhibit joint damage and permanent invalidity [13]. Psoriasis patients without PsA reportedly show substantial signs of enthesophyte formation [14]. Early psoriatic arthritis (ePsA), which is defined as inflammatory joint symptoms and signs compatible with PsA of less than 24 months of duration [13], usually appears as enthesoarthritis with a consistent risk of evolving toward erosive and deforming arthritis in the first year of disease [15,16]. Several studies recently demonstrated that the early diagnosis and intervention of PsA prevent permanent invalidity.

This review focuses on the current knowledge regarding the diagnosis of ePsA and discusses the significance of its early intervention.

## 2. Recent Concept of PsA Onset

It is difficult to determine when PsA begins in individual patients. PsA is usually diagnosed when patients present with psoriasis skin lesions and rheumatoid factor (RF)-negative inflammatory arthritis. Recent literature argues that the pathophysiology of PsA starts at a much earlier time point several years prior to the diagnosis of PsA. The Delphi consensus study proposed three stages for such patients as follows: (1) individuals with psoriasis at increased risk for PsA; (2) individuals with psoriasis and asymptomatic synovio-entheseal imaging abnormalities; and (3) individuals with psoriasis and musculoskeletal symptoms not explained by other diagnoses [17].

### 2.1. Individuals with Psoriasis with Increased Risk for PsA

Patients with psoriasis are at a higher risk of developing PsA than healthy controls and at a higher risk of developing PsA than other patients [6,18,19]. Thus, there is a keen need to identify psoriasis patients at a higher risk of developing PsA to prevent progression to PsA, but knowledge of this is limited.

Some clinical features, such as nail pitting and scalp and genital involvement, are predictors of PsA in patients [20]. Others include obesity, the presence of arthralgia, severe psoriasis, a history of uveitis, nail psoriasis, scalp psoriasis, having a first-degree relative with PsA, and any associated gene (such as human leukocyte antigens [HLA]-B*08, HLA-B*27, HLA-B*38, and HLA-B*39) [6,21].

### 2.2. Individuals with Psoriasis and Asymptomatic Synovio-Entheseal Imaging Abnormalities

Recent studies revealed that, in some patients with psoriasis, even those without symptoms of arthritis such as joint swelling or pain, imaging analysis with magnetic resonance imaging (MRI) or ultrasonography (US) demonstrate abnormalities [22,23]. These modalities include MRI for axial disease, MRI for peripheral arthritis, US for peripheral arthritis, US for enthesis, and plain radiography for peripheral arthritis [24].

### 2.3. Individuals with Psoriasis and Musculoskeletal Symptoms Not Explained by Other Diagnoses

Some patients with psoriasis report heel pain, stiffness, and/or arthralgia, which are not explained by other diagnoses, and no imaging abnormalities [25]. Previous studies identified these patients as “prodromal PsA”, “subclinical PsA”, “psoriasis with arthralgia”, “psoriasis with musculoskeletal symptoms”, or “psoriasis with musculoskeletal symptoms without musculoskeletal signs”.

The progression of these stages to PsA is shown in Figure 1.

More than 80% of PsA patients develop after the diagnosis of psoriasis (PsO), thus it is important to recognize PsO patients with increased risk to develop PsA. Nail psoriasis, scalp, and genital skin involvement are the risk factors to develop PsA in PsO patients. PsO patients with arthralgia (PsOAr) are also at higher risk to develop PsA. Almost 50% of PsO patients without articular symptoms show subclinical arthritis detected by imaging techniques. Among them, those who have active enthesitis detected by ultrasonography have higher risk to progress into PsA [27].

## 3. Questionnaires

Questionnaires that include key questions about joint symptoms, morning stiffness, and function can aid the diagnosis of ePsA. Three representative questionnaires are available for screening patients with psoriasis and arthritis. First, the Psoriatic Arthritis Screening and Evaluation (PASE) questionnaire is an effective screening tool for detecting patients with PsA [28]. It reportedly has a sensitivity of 82% and a specificity of 73% [28]. Second, the Psoriasis and Arthritis Questionnaire (PAQ), first reported in 1997, can predict PsA in patients with psoriasis with a sensitivity of 85% and specificity of 88% [29]. A validation study of the PAQ showed a sensitivity of 60% and specificity of 62% [29]. The modified PAQ features improved sensitivity and specificity of 68.7% and 77.8%, respectively [29].

Third, the Toronto PsA screening questionnaire (ToPAS) evaluates the clinical features of patients with PsA [30]. Its inclusion of pictures of skin and nail lesions distinguishes it from other screening questionnaires. Although PASE and PAQ are limited for detecting arthritis in patients with psoriasis, the ToPAS can screen for PsA regardless of whether a patient has psoriasis [30]. Its sensitivity and specificity are reportedly 94% and 92% [30]. The higher sensitivity and specificity can help identify patients with PsA in various clinical settings.

The psoriasis epidemiology screening tool (PEST) was first presented in 2009 by Ibrahim et al. and developed on a primary care–based population of psoriasis patients [31]. It consisted of five simple questions and had a sensitivity and specificity of 92% and 78%, respectively [31]. The Early Arthritis for Psoriatic Patients (EARP) questionnaire, developed by Tinazzi et al. in 2012, consists of 10 simple questions and features a sensitivity and specificity of 91% and 85%, respectively [32].

The Screening Tool for Rheumatologic Investigation in Psoriatic Patients (STRIPP) tool was recently developed by Burlando et al. [33] The STRIPP is composed of six sections. The first section is about demographic data, such as age, sex, psoriasis onset, and the second part evaluates psoriasis with PASI and specific localization such as the nails, scalp, and genitalia. The third part concerns ongoing treatment. The fourth section is derived from the PASE with six questions, the fifth section is about uveitis and inflammatory bowel diseases, and the sixth section focuses on rheumatological evaluation with imaging and diagnosis. The sensitivity and specificity are 91.5% and 93.3%, respectively [33].

Several reports on the comparison of these psoriatic arthritis screening tools have been published, some revealed similar efficacy, and others, comparing EARP, PEST, PASE, and toPAS II, revealed that EARP showed the highest sensitivity, and ToPAS II showed highest specificity. However, these tools present relatively low specificity, allowing other causes of musculoskeletal pain to be evaluated as PsA. This is because PsA is a heterogeneous entity and developing a screening tool to identify PsA and exclude other causes of musculoskeletal pain is extremely difficult. Thus, we have to be aware that we might seeing patients with other musculoskeletal disease than PsA, even when they are screened by questionnaires [34,35,36].

### Classification Criteria

Given the absence of a validated case definition of PsA, scientific and clinical research on PsA has been a major problem. However, an international group of rheumatologists proposed the Classification of Psoriatic Arthritis (CASPAR) criteria in 2006, which remain the current representative criteria based on the results of a large prospective study [37]. The CASPAR criteria were developed for use in clinical research and had a sensitivity and specificity of 91.4% and 98.7%, respectively, in patients with other forms of inflammatory arthritis. The high sensitivity and specificity suggest that it may also be used as a diagnostic criterion for PsA. Several studies have tested the sensitivity of CASPAR criteria for detecting early PsA. Classification criteria such as CASPAR are generally not useful for diagnostic purposes; however, their application for detecting ePsA remains to be established [37]. Since the initial development of CASPAR criteria, many studies have been conducted to establish its effectiveness as a criterium and also as a diagnostic tool, resulting in frequent use of CASPAR criteria in various clinical studies on PsA [38,39,40].

## 4. Biomarkers of ePsA

Biological markers (biomarkers) are objective and useful markers for the diagnosis and evaluation of alterations in physiological status [41]. To date, no disease-specific biomarkers have been identified for ePsA. PsA usually tests negative for rheumatoid factor, which differentiates it from rheumatoid arthritis (RA), the most common form of inflammatory arthritis [10]. However, many candidate biomarkers with potential utility in PsA have been reported [10], such as elevated erythrocyte sedimentation rate (ESR), C-reactive protein (CRP), and acute-phase serum amyloid A (A-SAA), all of which are nonspecific inflammatory markers that are also elevated in RA. Some cytokines are elevated in synovial fluids in PsA with polyarticular involvement compared to those with monoarticular involvement, such as interleukin (IL)-1, IL-12p40, interferon alpha, IL-15, and chemokine ligand 3, which could differentiate PsA patients with polyarticular involvement from those with oligoarticular involvement [10].

S100A8/S100A9 (calgranulin) levels are elevated in patients with high disease activity, which is decreased by treatment with methotrexate with a decreased number of swollen joints, the Richie articular index, and a disease activity score [42]. They are elevated in patients with >10 involved joints compared to those with <10 involved joints [43]. Vascular endothelial growth factor and angiopoetin-2 are angiogenic markers that predict joint damage in RA, and their levels are higher in PsA than in RA, which also predicts joint damage in PsA [43]. The radiographic progression of PsA patients correlates with the levels of macrophage colony-stimulating factor and receptor activator of nuclear factor kappa B ligand [44]. Baseline levels of A-SAA correlated with 1-year radiographic progression in patients with PsA. A-SAA levels correlate with the levels of matrix metalloproteinase (MMP)1, MMP3, MMP13, and tissue inhibitors of matrix metalloproteinases [45]. A-SAA is known to induce MMP production in synovial fibroblast-like cells [46], and MMP1 and MMP3 are reportedly associated with radiographic progression in patients with early RA [47], which suggests that they could also be early disease progression markers for PsA.

Some genetic markers indicate psoriasis and PsA as distinct populations. Many molecules have demonstrated differences in the prevalence of psoriasis and PsA, but most, even if involved in the pathophysiology, may not be involved in the pathogenesis, showing only very low correlation with the disease. HLA molecules are the only molecules that have been identified as risk factors for PsA [6,21].

Currently unidentified epigenetic markers can be used to distinguish psoriasis from PsA. IL-22 is one such candidate whose methylation levels in patients with cutaneous symptoms only and those with cutaneous and articular involvement changes [48].

## 5. Imaging Techniques

Imaging techniques are more sensitive than clinical examinations in the diagnosis of synovitis and enthesitis as well as the assessment of inflammatory activity in PsA. Incorporating imaging modalities in the assessment and early intervention of ePsA may be useful for preventing permanent invalidity. In early PsA, inflammatory changes occur in the soft tissue and bone marrow that cannot be detected using plain X-rays [9]. Ultrasonography and MRI are sensitive and useful tools for detecting inflammatory joint disease [49,50,51]. Ultrasonography is frequently used to evaluate arthritis. Recent studies by Zabotti et al. [52] revealed that psoriasis patients with arthralgia (PsOAr) are at higher risk to develop PsA, with higher positive sonographic findings of tenosynovitis, which was not correlated with development to PsA in longitudinal study. Sonographically determined active enthesitis was associated with disease progression to PsA.

Synovio-entheseal complex (SEC) has been shown the initial site of inflammation in PsA, where mechanistic stress occurs, which efficiently distinguishes PsA from RA. It has been revealed that up to half of asymptomatic psoriasis patients showed subclinical synovial or entheseal inflammation [27]. The most important findings suggestive of early PsA in ultrasonography is enthesitis of metacarpophalangeal (MP) joints, and proximal interphalangeal (PIP) joints of the hands [52]. The diagnosis of axial disease might require MRI, whereas enthesitis can be visualized using both MRI and ultrasound (US) [24]. Dynamic MRI may be a clinically useful measure of synovial inflammation. High-resolution peripheral quantitative computed tomography (HRpQ-CT) is a novel technique mainly used for the diagnosis and evaluation of disease progression [53]. High-resolution fluorine-18 fludeoxyglucose (^18^F-FDG) positron emission tomography (PET)/CT imaging of the wrist and hand is feasible in RA or PsA patient cohorts and can provide quantifiable measures of disease activity [54,55]. In addition, it has been reported that ^18^F-FDG) PET/CT is a powerful tool for detecting subclinical arthritis in patients with psoriatic arthritis and/or psoriasis vulgaris [56].

## 6. Treatment of ePsA

The clinical signs and symptoms of ePsA often fluctuate, and the disease course is not simply in one direction; rather, it moves back and forth. Some sPsA patients rapidly progress to severe disease, whereas other ePsA patients develop clinical symptoms that disappear over time. Thus, our understanding is that some ePsA patients require early intervention to prevent the development of severe disease but others do not because their disease remains mild or spontaneously improves. If good markers were available to distinguish between patients with versus without severe ePsA severe disease, it would be easy to treat these patients. However, there are no such markers; therefore, we treat patients according to their current disease severity.

### 6.1. Nonsteroidal Anti-Inflammatory Drugs and Methotrexate

The first medications we tried for patients with ePsA were non-steroidal anti-inflammatory drugs (NSAIDs). Some patients are sufficiently treated with NSAIDs and the remaining mild disease or symptoms disappear during the disease course [57].

Other patients require more efficient treatment such as methotrexate (MTX) [57]. MTX is approved for use in severe psoriasis (which is often related to psoriatic arthritis) and rheumatoid arthritis (RA) by the U.S. Food and Drug Administration (FDA) and the European Medicines Agency (EMA), and frequently used to treat PsA, in spite of the lack of evidence with randomized controlled trials (RCTs). Pincus T et al. [58] discussed on MTX clinical trials on PsA and suggested that too high or too low dose of MTX use, insufficient stratification of patients, or insufficient statistical power to detect differences in old clinical trials caused the lack of evidence of MTX, and that the treatment advantage versus placebo without statistical significance (*p* < 0.05) does not necessarily mean the absence of clinical efficacy.

MTX was intensely used to treat PsA patients before biologics emerged, although it was not approved in Japan before March 2019 [59]. Adverse effects, including liver toxicity and hematopoiesis suppression, are disadvantages of this drug.

Leflunomide, a selective pyrimidine synthesis inhibitor with the property to inhibit T-cell activation and proliferation has also been shown to improve joint and skin symptoms of PsA (although with less efficacy in the skin) [57]. Leflunomide has been shown effective in several randomized double-blind placebo-controlled studies in PsA, but MTX has not [60,61].

The European League Against Rheumatism (EULAR) recommendation for the management of psoriatic arthritis recommends conventional synthetic disease-modifying antirheumatic drugs (csDMARDs) such as MTX or leflunomide for peripheral arthritis with polyarticular involvement, monoarthritis, or oligoarthritis with poor prognostic factors such as structural damage, high ESR and CRP levels, dactylitis, or nail involvement [57]. Biologic DMARDs (bDMARDs) are recommended when csDMARDs are ineffective.

### 6.2. Biologics

Tumor necrosis factor (TNF) antagonists are well established for the treatment of PsA [57]. To date, four anti-TNF agents, infliximab, adalimumab, and certolizumab-pegol, have been approved for the treatment of PsA by the Japanese authorities [62]. These agents effectively improve articular symptoms of peripheral and axial diseases, radiological findings, and skin and nail lesions [63].

Anti-IL-17 antibodies, including secukinumab, ixekizumab, and brodalumab, have proven effective at treating PsA with peripheral and axial involvement. The EULAR recommendation for the treatment of PsA recommends TNF inhibitors and IL-17 inhibitors for axial disease as the first choice because MTX and IL-23 are inferior [57]. On the other hand, IL-12/23 antibodies are effective for peripheral arthritis and are recommended at the same level as IL-17 and superior to TNF inhibitors for peripheral arthritis when the csDMARD efficacy is inadequate [57].

Anti-IL-23 antibodies including guselkumab, risankizumab, and tildrakizumab, have less efficacy for treating axial disease compared to anti-IL-17 antibodies and anti-TNF antibodies, maybe because IL-17-producing cells are independent on IL-23 stimulation in axial lesion, such as ɤδ T cells and mucosal-associated invariant cells [64]. Radiologic investigations, including MRI and HRpQ-CT, showed the absence of both erosive and bone anabolic damage, supporting the possibility of the arrest of progression of anabolic changes in PsA with secukinumab and ixekizumab.

Treatment of psoriasis and psoriatic arthritis with biologics have been shown to protect patients from systemic inflammatory comorbidities, such as cardiovascular diseases, diabetes, and abnormal lipid metabolism. It needs caution in that anti-IL-17 antibodies may cause newly onset or worsening of inflammatory bowel diseases [63].

### 6.3. Janus Kinase Inhibitors

Janus kinase (JAK) inhibitors were recently approved for the treatment of PsA in Japan. JAK inhibitors are classified as target-specific DMARDs (tsDMARDs). Owing to the adverse effects of this category of drugs, they are recommended for the treatment of PsA when bDMARDs are ineffective. The efficacy and safety of JAK inhibitors for PsA were recently discussed. Three JAK inhibitors–tofacitinib, baricitinib, and upadacitinib–have been approved for use in autoimmune diseases; of them, only tofacitinib has been approved for the treatment of PsA. Tofacitinib, an orally available JAK inhibitor, broadens the treatment options for PsA and other inflammatory conditions [65].

## 7. Early Intervention for ePsA

However, when to implement early intervention for PsA remains controversial. The open prospective exploratory Interception in Very Early PsA (IVEPSA) study showed that very early disease intervention with secukinumab, an IL-17A inhibitor, for PsA may lead to a comprehensive decline in skin symptoms [66]. The Tight Control Of inflammation in early Psoriatic Arthritis (TICOPA) study showed the effects in patients in the tight control group [67]. Moreover, trials to demonstrate the efficacy of targeted biologic therapies and DMARDs in early PsA will test the validity of early intervention as a strategy to alter the disease course [67].

Biologic treatment of psoriasis patients without psoriatic arthritis have reported to reduce the incidence of development of psoriatic arthritis [68]. Psoriasis patients without psoriatic arthritis may include those with increased risk, or with asymptomatic arthritis with imaging abnormalities, or with undiagnosed musculoskeletal symptoms as discussed in Section 2. It would be of importance to identify patients in need to be treated with bDMARDs to prevent the development of psoriatic arthritis, to avoid overtreatment.

However, the disease course of PsA is not simple, and various patients follow distinct disease courses with an ever-expanding and fluctuating disease course. Novel biomarkers that distinguish patients who need early intervention are needed to fully prevent disease progression in those with a poor prognosis.

### 7.1. Guidelines

The Group for Research and Assessement of Psoriasis and Psoriatic Arthritis (GRAPPA), European League Against Rheumatism (EULAR), American college of Rheumatology/National Psoriasis Foundation, and other national associations of dermatologists in each country including Japan, Great Britain, Germany, and so on, have published guidelines for treating psoriasis and psoriatic arthritis, and continue updating them to include the most recent advancements in treatment of psoriasis and psoriatic arthritis [69,70,71,72]. GRAPPA is a global research group for psoriasis and psoriatic arthritis, assessing both dermatological and musculoskeletal manifestations, while EULAR focuses on rheumatic diseases referring to dermatologists for significant skin disease but not recommendations for skin and nail manifestations. Each country has its own system of insurance and it is hard to establish a general recommendation to fit systems in all countries, and each country establishes its own guidelines referring to and modifying EULAR and/or GRAPPA. EULAR bases on Oxford Center for Evidence-Based Medicine: Levels of Evidence, and GRAPPA relies on newer Grading of Recommendations, Assessment, Development and Evaluation [73]. Both of them are based on a specific systematic literature review (SLR). GRAPPA mostly depends on randomized control trials. However, good RCTs are sometimes missing, especially for those medicines from old time, such as methotrexate (MTX). EULAR recommend MTX as a first line in treatment for PsA, based on experts’ opinions, while GRAPPA dose not give rank to MTX, although it is included as one of potential DMARDs.

### 7.2. Costs

Although biologics are quite effective for the manifestation and health-related quality of life of PsA, they may increase the economic burden on health systems [74]. The total annual cost per patient ranged from US $10,924 to US $17,050, with purchasing power parity for PsA in five European countries [75]. It has also been reported that the introduction of biologics leads to a 3-fold to 5-fold increase in direct costs and, consequently, an increase in total costs [76].

The EULAR recommendation and Japanese guideline for the treatment of PsA do not recommend biologics as the first-choice treatment of PsA; rather, they recommend csDMARDs or MTX [57,76]. Biologics are highly efficient drugs for treating PsA, but their cost would burden the country’s economy. In contrast, csDMARDs including MTX are inexpensive and effective drugs for the treatment of PsA and should be used before bDMARDs in these countries. However, the American College of Rheumatology/National Psoriasis Foundation guidelines recommend biologics at the same level as csDMARDs due to the different insurance systems, mostly dependent on private insurance companies [77]. Each country adopts EULAR and GRAPPA recommendations, modifying them to fit its insurance system. Because recently developed biologics and targeted therapeutics cost tremendously, in some countries, the use of them are restricted to certain period of time. It would be necessary to develop guidelines to benefit both patients and social insurance systems effectively to continue providing good medical treatments.

Even with insurance, some patients cannot afford biologics for PsA treatment. There are certainly economic disparities in modern society in which many patients are not adequately treated because of economic reasons.

## 8. Conclusions

Despite tremendous advances in therapies and treatment strategies, there remains an unmet need to identify the optimal therapeutic approach for individual PsA patients. The diagnosis and intervention of ePsA are important to preventing disease progression, structural damage, and permanent invalidity. Therefore, standardized imaging techniques, validated scoring systems, and protocols are required. New imaging techniques such as US, MRI, and PET/CT have since been developed. Despite these developments, there is currently no gold standard technique to detect ePsA. Psoriatic arthritis is a heterogenous condition, which includes preclinical, subclinical, mild to severe disease, and these conditions may or may not progress to severer condition, depending on individual cases, which makes it difficult to establish a simple guidance to apply to all patients. Early intervention for PsA will probably inhibit inflammation and alter the disease course, and it is of importance to distinguish patients in need for early intervention not to overtreat mild disease patients and to save costs. However, an efficient tool to distinguish such patients in need and the evidence to support this is still lacking. Thus, further studies on the pathophysiology, diagnosis and intervention of ePsA are required.

## Figures and Tables

**Figure 1 jcm-11-02051-f001:**
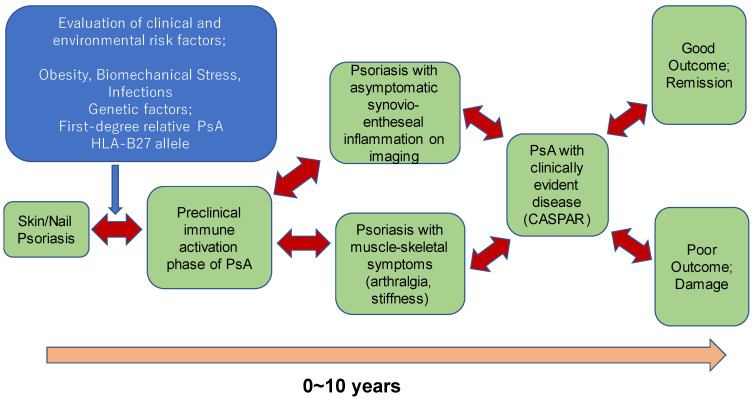
The natural clinical course of PsA, including its preclinical stages. Each stage can be reversed as represented by the two-way arrows. Adapted from Pennington and Fitzgerald [26], Frontiers in Medicine 2021 with permission.

## Data Availability

Not applicable.

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
