# Peer review of "Diagnosis and Intervention in Early Psoriatic Arthritis"

_jcm, 2022, doi:10.3390/jcm11072051_

Round 1

Reviewer 1 Report

Diagnosis and Intervention in Early Psoriatic Arthritis

I presume this is an invited review. The authors need to be more cautious in their recommendations/conclusions – the evidence they quote for early diagnosis and intervention is purely observational and should be regarded as such. Psoriatic arthritis is such an heterogeneous condition that there are no ‘simple rules’ to guide the clinician and this should be emphasised in the article.

Screening questionnaires are useful but limited in sensitivity and specificity. The authors quote several but make no comment or do not allude to comparative studies, or how they may be used more effectively in primary care of by dermatologists.

The CASPAR criteria are classification criteria, developed with scientific rigour, and have been shown to work well as diagnostic criteria and in early disease – these studies should be quoted.

Finally, to be balanced the authors should refer to other international recommendations, such as those from GRAPPA, and how these might be implemented in countries where funds are not freely available.

Author Response

To reviewer 1:

Thank you very much for your time to review our manuscript.

We revised our manuscript following your comments.

I presume this is an invited review. The authors need to be more cautious in their recommendations/conclusions – the evidence they quote for early diagnosis and intervention is purely observational and should be regarded as such. Psoriatic arthritis is such a heterogeneous condition that there are no ‘simple rules’ to guide the clinician and this should be emphasised in the article.

Response: We added some sentences to emphasize that psoriatic arthritis is heterogeneous condition and there are no simple rules to guide the clinician as follows.

This is because PsA is a heterogeneous entity, and developing a screening tool to identify PsA and exclude other causes of musculoskeletal pain is extremely difficult. Thus, we have to be aware that we might seeing patients with other musculoskeletal disease than PsA, even when they are screened by questionnaires [33-35]. (Line 127-131)

Psoriatic arthritis is a heterogenous condition, which includes preclinical, subclinical, mild to severe disease, and these conditions may or may not progress to severer condition, depending on individual cases, which makes it difficult to establish a simple guidance to apply to all patients. Early intervention for PsA will probably inhibit inflammation and alter the disease course, and it is of importance to distinguish patients in need for early intervention not to overtreat mild disease patients and to save costs. However, efficient tool to distinguish such patients in need and the evidence to support this is still lacking. Thus, further studies on the pathophysiology, diagnosis and intervention of ePsA are required. (Line 351-360)

Screening questionnaires are useful but limited in sensitivity and specificity. The authors quote several but make no comment or do not allude to comparative studies, or how they may be used more effectively in primary care of by dermatologists.

Response: We added some literature comparing several questionnaires and comments as follows.

Several reports on comparison of these psoriatic arthritis screening tools have been published, some revealed similar efficacy, and others, comparing EARP, PEST, PASE, and toPAS II, revealed that EARP showed the highest sensitivity, and ToPAS II showed highest specificity. However, these tools present relatively low specificity, allowing other causes of musculoskeletal pain to be evaluated as PsA. This is because PsA is a heterogeneous entity, and developing a screening tool to identify PsA and exclude other causes of musculoskeletal pain is extremely difficult. Thus, we have to be aware that we might seeing patients with other musculoskeletal disease than PsA, even when they are screened by questionnaires [33-35]. (Line 123-131)

The CASPAR criteria are classification criteria, developed with scientific rigour, and have been shown to work well as diagnostic criteria and in early disease – these studies should be quoted.

Response: We added literatures on studies on CASPAR criteria as follows.

Since the initial development of CASPAR criteria, many studies have been conducted to establish its effectiveness as a criteria and also as a diagnostic tool, resulting in frequent use of CASPAR criteria in various clinical studies on PsA [37-39]. (Line 143-145)

Finally, to be balanced the authors should refer to other international recommendations, such as those from GRAPPA, and how these might be implemented in countries where funds are not freely available.

Response: We added sentences in the text refering to other international recommendations as follows.

Group for Research and Assessement of Psoriasis and Psoriatic Arthritis (GRAPPA), European League Against Rheumatism (EULAR), American college of Rheumatology/National Psoriasis Foundation, and other national association of dermatologists in each country including Japan, Great Britain, Germany, and so on, have published guidelines for treating psoriasis and psoriatic arthritis, and continue updating them to include the most recent advancements in treatment of psoriasis and psoriatic arthritis [70-73]. GRAPPA is a global research group for psoriasis and psoriatic arthritis, assessing both dermatological and musculoskeletal manifestations, while EULAR focuses on rheumatic diseases referring to dermatologists for significant skin disease but not recommendations for skin and nail manifestations. Each country has its own system of insurance, it is hard to establish a general recommendations to fit systems in all countries, and each country establishes its own guidelines referring to and modifying EULAR and/or GRAPPA. EULAR bases on Oxford Center for Evidence-based Medicine: Levels of Evidence, and GRAPPA relies on newer Grading of Recommendations, Assessement, Development and Evaluation [74]. Both of them are based on a specific systematic literature review (SLR). GRAPPA mostly depends on randomized control trials. However, good RCTs are sometimes missing, especially for those medicines from old time, such as methotrexate (MTX). EULAR recommend MTX as a first line in treatment for PsA, based on experts’ opinions, while GRAPPA dose not give rank to MTX, although it is included as one of potential DMARDs. (line 300-319)

Reviewer 2 Report

In this review, authors discuss the most recent literature concerning early psoriatic arthritis. This is a relevant topic and of great interest to rheumatologists.

The review is well written and addresses the most important evidence. However, I recommend to deeply discuss some points:

  • Some recent studies have evaluated the role of bDMARDs in the development of PsA among patients with psoriasis. This can be discussed in page 2, section 2.1.

  • In section 5, it would be interesting to explain the most important findings suggestive of PsA in ultrasonography.

  • In section 6.1, It should be explained that MTX did not demonstrate efficacy in PsA, in contrast to leflunomide did. Leflunomide as a treatment in PsA should be reflected in the text.

  • Since this manuscript will have an international audience, authors should also refer to drugs approved by the FDA and EMA, not only drugs approved in Japan.

Minor comments

  • In the introduction (line 22), “peripheral arthritis” and “asymetrical synovitis” is redundant.

  • Please review the following sentence (line 32): “…and intervention with a rapid therapeutic intervention…”

  • In section 3.1, the first and second paragraph are the same. Please remove one of them.

Author Response

To reviewer 2:

Thank you very much for your time to review our manuscript. We revised our manuscript following your comments.

In this review, authors discuss the most recent literature concerning early psoriatic arthritis. This is a relevant topic and of great interest to rheumatologists.

Response: Thank you for your favorable comments.

The review is well written and addresses the most important evidence. However, I recommend to deeply discuss some points:

  • Some recent studies have evaluated the role of bDMARDs in the development of PsA among patients with psoriasis. This can be discussed in page 2, section 2.1.

Response: We added recent studies to evaluate the role of bDMARDs in the development of PsA among patients with psoriasis, in section 7, referring to the classification in section 2.1. as follows.

Biologic treatment of psoriasis patients without psoriatic arthritis have reported to reduce the incidence of development of psoriatic arthritis [69] Psoriasis patients without psoriatic arthritis may include those with increased risk, or with asymptomatic arthritis with imaging abnormalities, or with undiagnosed musculoskeletal symptoms as discussed in section 2. It would be of importance to identify patients in need to be treated with bDMARDs to prevent the development of psoriatic arthritis, to avoid overtreatment. (Line 289-294)

  • In section 5, it would be interesting to explain the most important findings suggestive of PsA in ultrasonography.

Response: We added the literature on ultrasonographic findings of early PsA, and mentioned on it as follows.

Ultrasonography is frequently used to evaluate arthritis. Recent studies by Zabotti et al [51 revealed that psoriasis patients with arthralgia (PsOAr) are at higher risk to develop PsA, with higher positive sonographic findings of tenosynovitis, which was not correlated with development to PsA in longitudinal study. Sonographically determined active enthesitis was associated with disease progression to PsA. Synovio-entheseal complex (SEC) has been shown the initial site of inflammation in PsA, where mechanistic stress occurs, which efficiently distinguishes PsA from RA. It has been revealed that up to half of asymptomatic psoriasis patients showed subclinical synovial or entheseal inflammation [52]. The most important findings suggestive of early PsA in ultrasonography is enthesitis of metacarpophalangeal (MP) joints, and proximal interphalangeal (PIP) joints of the hands [51]. (Line 189-199)

  • In section 6.1, It should be explained that MTX did not demonstrate efficacy in PsA, in contrast to leflunomide did. Leflunomide as a treatment in PsA should be reflected in the text.

Response: We added the statement on MTX and leflunomide in section 6.1 as follows.

MTX is approved for use in severe psoriasis (which is often related to psoriatic arthritis) and rheumatoid arthritis (RA) by the U.S. Food and Drug Administration (FDA) and the European Medicines Agency (EMA), and frequently used to treat PsA, in spite of the lack of evidence with randomized controlled trials (RCTs). Pincus T et al. [58] discussed on MTX clinical trials on PsA and suggested that too high or too low dose of MTX use, insufficient stratification of patients, or insufficient statistical power to detect differences in old clinical trials caused the lack of evidence of MTX, and that the treatment advantage versus placebo without statistical significance (P<0.05) does not necessarily means the absence of clinical efficacy. (Line 224-232)

Leflunomide, a selective pyrimidine synthesis inhibitor with the property to inhibit T-cell activation and proliferation has also been shown to improve joint and skin symptoms of PsA (although with less efficacy in the skin) [57]. Leflunomide has been shown effective in several randomized double-blind placebo-controlled studies in PsA, but MTX has not [60,61]. (Line 236-240)

  • Since this manuscript will have an international audience, authors should also refer to drugs approved by the FDA and EMA, not only drugs approved in Japan.

Response: We added the statement on FDA and EMA as well as Japan as follows.

MTX is approved for use in severe psoriasis (which is often related to psoriatic arthritis) and rheumatoid arthritis (RA) by the U.S. Food and Drug Administration (FDA) and the European Medicines Agency (EMA), and frequently used to treat PsA, in spite of the lack of evidence with randomized controlled trials (RCTs). (Line 224-227)

Minor comments

  • In the introduction (line 22), “peripheral arthritis” and “asymetrical synovitis” is redundant.

Response: We have deleted these words.

  • Please review the following sentence (line 32): “…and intervention with a rapid therapeutic intervention…”

Response: We have deleted the redundant word.

  • In section 3.1, the first and second paragraph are the same. Please remove one of them.

Response: We have deleted the paragraph.

Round 2

Reviewer 1 Report

none

Reviewer 2 Report

My comments have been well addressed.